

# Three dominant synoptic atmospheric circulation patterns influencing severe winter haze in eastern China

Shiyue Zhang[1], Gang Zeng[1]*, Tijian Wang[2], Xiaoye Yang[1], Vedaste Iyakaremye[3]

[1]Key Laboratory of Meteorological Disaster, Ministry of Education, Collaborative Innovation Center on Forecast and Evaluation of Meteorological Disasters (CIC-FEMD), Joint International Research Laboratory of Climate and Environment Change (ILCEC), Nanjing University of Information Science and Technology, Nanjing, China, 210044

[2]School of Atmospheric Sciences, Nanjing University, Nanjing, 210023, China

[3]Rwanda Meteorology Agency, Nyarugenge KN 96 St, Kigali, Rwanda

*Correspondence to: Gang Zeng (zenggang@nuist.edu.cn)*

**Abstract.** Previous studies indicated that, on synoptic scale, the severe haze in eastern China (EC) is affected by the atmospheric circulation variations. However, it is still unclear what are the dominant atmospheric circulation patterns influencing the severe winter haze conditions in EC and what are the differences between them. To systematically determine the dominant synoptic atmospheric circulation patterns of severe haze in different regions of EC, we use the Hierarchical Clustering Algorithm to classify the local geopotential height anomalies at 500-hPa over the stations with severe haze and obtained three dominant synoptic atmospheric circulation types based on observed $PM_{2.5}$ concentration and NCEP/NCAR reanalysis. Circulation Type1 is accompanied by significant north wind component anomalies over northern China and causes severe haze pollution over the Yangtze River valley. Although the local meteorological conditions are not conducive to haze formation and accumulation, the severe haze in Yangtze River valley is related to the pollution transportation caused by the north wind anomalies. During the haze days with circulation Type2, the joint affection of East Atlantic-West Russia teleconnection pattern and winter East Asia subtropical jet stimulate and maintain the anticyclonic anomalies over northeast Asia, which provides meteorological conditions conducive to the occurrence of severe haze over the whole EC. The circulation Type3 mainly caused severe haze events in northeast China through the establishment of blocking high over the Okhotsk Sea. The results provide a basis for establishing haze prediction and management policies applicable to different regions in EC.



## 1. Introduction


Severe haze could increase the risk of traffic accidents by reducing visibility and harm human
health by causing respiratory diseases (Xie et al., 2014; Hu et al., 2015; Wang et al., 2016). Haze
events in China are mainly caused by $PM_{2.5}$ (particulate with an aerodynamic diameter less than
2.5μm; Cai et al., 2017; Shen et al., 2018; Wang et al., 2021). Researches show that the distribution
of haze days in China has the characteristics of uneven spatial distribution, with more in
economically developed eastern region and less in economically underdeveloped region (Wu et al.,
2013; Liu et al., 2015; Xu et al., 2015). With the rapid development of industrialization, urbanization
and increase in anthropogenic emission, eastern China (EC) has experienced more severe haze
events with long duration, large spatial scale, and serious harm in the past few decades (Monks et
al., 2009; Qian et al., 2009; Wang et al., 2009). Since the beginning of the 21st century, the uneven
spatial distribution of haze events in China have become more obvious (Sun et al., 2016), which has
led to the increasing incidence rate and mortality related to respiratory diseases in Beijing-Tianjin-
Hebei, the Yangtze River valley, and the Pearl River Delta (Tsaia et al., 2014; Ding et al., 2016; Fan
et al., 2019). Although haze pollution control in China has been improved to some extent with the
strict implementation of energy conservation and emission reduction policies after 2013 (Wang et
al., 2021), it still affects various socio-economic sectors and human health.
In addition to human activities, meteorological conditions are also considered as one of the most
important factors for determining regional air quality. Previous studies have indicated that, on the
weather scale, the formation and maintenance of haze days in eastern China ($HD_{EC}$) are closely
related to favorable weather conditions (Niu et al., 2010; Cai et al., 2017), including strong thermal
inversion potential, high relative humidity, negative sea level pressure anomaly, and weak wind
speed. Furthermore, the anticyclonic anomaly could lead to the sinking movement and weaker
thermal inversion potential, which inhibit the vertical diffusion of pollutants and affect the air quality
of the local or larger region (Wu et al., 2013; Xu et al., 2015). Many studies investigated the key
circulation system affecting $HD_{EC}$ from an interannual scale or intraseasonal scale and suggested
that the weak East Asian Winter Monsoon (Li et al., 2015; Yin et al., 2015; Zhang et al., 2022), the
positive phase of Arctic Oscillation (Wang et al., 2015; Yin et al., 2015) and the positive phase of
East Atlantic-West Russia (EA/WR) teleconnection pattern (Yin et al., 2017) could result in more





haze days in China. On the synoptic scale, meteorological conditions could also significantly
regulate HD_{EC}. The weak synoptic circulation with a high-pressure or continuous low-pressure
system is beneficial for the accumulation of pollution, while the strong weather phenomena with a
large pressure gradient encourage the diffusion of pollutants (Li et al., 2019; Cai et al., 2020).
Furthermore, studies have shown that cold surges can dissipate and reduce local air pollutants by
bringing dry and clean cold air (Wu et al., 2017; Leung et al., 2018; Zhang et al., 2021).

A recent study classified the daily winter circulation anomalies and suggested that there are two

dominant climate drivers (i.e., EA/WR teleconnection pattern and Victoria mode of sea surface
temperature anomalies) conducive to the severe haze occurrence in North China (Li et al., 2022).
However, there is still a lack of research on the dominant circulation patterns of severe HD_{EC}.
Therefore, the present study addresses the following scientific questions: (1) what are the synoptic
atmospheric circulation patterns that dominate severe haze pollution in EC, (2) what are the
differences in the action range of each circulation pattern, and what are their possible mechanisms.
These issues are addressed using a modified classification algorithm (Hierarchical Clustering
Algorithm) that is more suitable for studying the classification of synoptic patterns in a large spatial
range.

The remaining sections of this paper are structured as follows: Data and definitions are

introduced in section 2. Section 3 shows the dominant synoptic circulation patterns of severe HD_{EC}.
In section 4, we compare different circulation types associated with severe HD_{EC}. Finally, the
discussion and main conclusions are given in section 5.
**2.   Data and Methods**
**2.1   Data**

In this study, the daily meteorological data and the observed PM$_{2.5}$ concentrations from 2014 to

2021 were used to analyze the dominant circulation patterns and their main causes of severe haze
in winter in EC. The daily NCEP/NCAR reanalysis was obtained from https://psl.noaa.gov/, which
includes sea level pressure (SLP), surface air temperature (SAT), the temperature in multiple
pressure levels, geopotential height (GPH), three-dimensional wind, relative humidity (RH) at 1000-





hPa, and vertical velocity (omega) at 850-hPa (Kalnay et al., 1996). The dataset has a horizontal
resolution of 2.5°× 2.5°. In this study, we defined the thermal inversion potential (TIP) as the air
temperature at 850-Pa minus SAT referring to Yin and Wang (2019). The Daily $PM_{2.5}$ concentrations
for 935 meteorological stations in China (following Yin and Wang (2016) and Yin et al. (2021), the
stations with missing data more than 5% of are dropped; the stations with data lost continuously for
3 days or more is also discarded) were obtained from China National Environmental Monitoring
Centre (https://quotsoft.net/air/). The sporadic missing data (less than 3 days) were filled by cubic
spline interpolation.
**2.2  Definition of severe $HD_{EC}$**
In this study, the severe $HD_{EC}$ is defined when $PM_{2.5}$ concentration ≥ 150 µg m$^{-3}$ (Cai et al., 2017;
Zhong et al., 2019). We focused on the haze days in the cool season (November to February of the
following year, abbreviated as NDJF), which accounts for more than 40% of the total haze days in
China in a year (Sun et al., 2013; Wang et al., 2015). Figure 1 shows the climatology of haze days
in China from 2014 to 2021 in NDJF. The severe haze days are mainly concentrated in the EC (east
of 105°E and south of 54 °N), which is selected as the target area in the present study. Thus, a subset
of 853 stations is selected.

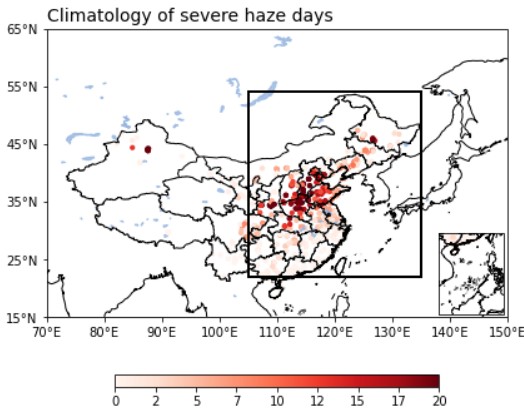


**Figure 1.** Spatial distribution of the annual averaged severe haze days (unit: day) in China from
2014 to 2021 in NDJF. The black box represents EC.





**2.3 Definition of blocking index**
In winter, the anticyclone anomaly over the Okhotsk Sea, usually related to atmospheric
blocking, may lead to haze accumulation (Yun and Yoo 2019; Hwang et al., 2022). Thus, based on
previous studies (Tibaldi et al., 1990; Fang and Lu, 2020), here we identify the blockings by
northward gradients (GHGN) and southward gradients (GHGS) of $Z_{500}$ at each grid point:
$$GHGN = \frac{z_{500}(\lambda,\phi+\Delta\phi)-z_{500}(\lambda,\phi)}{\Delta\phi} \qquad (1)$$
$$GHGS = \frac{z_{500}(\lambda,\phi)-z_{500}(\lambda,\phi-\Delta\phi)}{\Delta\phi} \qquad (2)$$
Where $\phi$ = 35°, 35.5° …, 75°N, $\lambda$ = 70°, 70.5° …, 160°E and $\Delta\phi$ = 15°. A given longitude is
defined as "blocked" at a particular time satisfies the following conditions:
$$GHGS > 0, \ GHGN < -10 \text{ m (deg lat)}^{-1}$$
Based on these conditions, we can identify whether any grid in the range of 35°N-70°N is blocked
at any time.
**2.4 Plumb's wave activity flux**
Here we used the wave flux of Rossby to show the propagation of wave energy (Plumb, 1985).
The two-dimensional Plumb's wave activity flux can be expressed by:
$$F_s = \frac{P}{P_0}\cos\varphi \times \begin{pmatrix} v'^2 - \frac{1}{2\Omega \, a \sin 2\varphi}\frac{\partial(v'\phi')}{\partial\lambda} \\ -u'v' + \frac{1}{2\Omega \, a \sin 2\varphi}\frac{\partial(u'\phi')}{\partial\lambda} \end{pmatrix} \qquad (3)$$
In Eq. (3), $F_s$ (unit: m$^{-2}$ s$^{-2}$) denotes the horizontal stationary wave activity flux, $P$ means the
pressure; $P_0$ =1000-hPa, $u'$ and $v'$ are the zonal and meridional wind deviation, respectively.
And the $\phi'$ is geopotential height. φ (λ) represents the latitude (longitude). a is the radius of Earth,
and $\Omega$ means Earth's rotation rate.

**2.5 Classification algorithm of synoptic atmospheric circulation**
This paper uses the hierarchical clustering algorithm (HCA) to classify the severe $HD_{EC}$ based
on the associated circulations anomalies. Based on HCA (Rokach et al., 2005), we could create a
clustering tree of data samples by calculating the Euclidean distance between different categories.



The original data samples of different types are at the lowest level of the tree, and the root point of
a cluster is at the top level of the tree.
Unlike Li et al. (2022), we only cluster the circulation anomalies of days with severe HD$_{EC}$,
which can ensure that all classification samples lead to PM$_{2.5}$ at least one station in EC exceeds the
standard of severe haze pollution and produce more accurate classification types. Secondly, the
circulation samples selected are not in fixed region, but the rectangular regions of the same size
centered on each station with severe haze (the GPH anomalies at 500-hPa in a rectangular region of
30 degrees from east, west, north, and south with each station as the center on the day of severe
HD$_{EC}$ were taken as the samples to perform HCA). It means that our classification results are
focused on the local circulation anomalies accompanied by haze.
We use the silhouette coefficient to determine the optimal classification result (Rousseeuw,
1987). For any sample $i$, the silhouette coefficient $s(i)$ is defined as:
$$s(i) = \frac{b(i)-a(i)}{max\{a(i),b(i)\}} \qquad (4)$$
$a(i)$ means the average distance from sample $i$ to all other samples in the cluster it belongs to,
and $b(i)$ means the lowest average distance from sample $i$ to all samples in any other cluster. The
silhouette coefficient of the clustering result is the average of the silhouette coefficients of all
samples. The closer to 1, the better the classification results. Figure S1 shows the clustering tree and
its associated silhouette coefficient of this study.
**3.  Dominant synoptic atmospheric circulation patterns of severe HD$_{EC}$**
Figure 2a shows the composite anomalies of 500-hPa GPH during all severe HD$_{EC}$ in 853
stations. Generally, the stations with severe haze are located in the southwestern parts of the
anticyclonic anomaly center, which is consistent with previous studies (Zhong et al., 2019; Wang
and Zhang, 2020). Then we performed the HCA as described in Section 2.5 and obtained three types
of dominant local circulation anomalies associated with the severe HD$_{EC}$ (Figure 2b, c, d).
Circulation Type1 shows a wave-train structure of '+ - +', and the stations are located in the west
of anticyclonic anomaly and the south of cyclonic anomaly. Circulation Type2 shows the circulation
anomalies similar to Figure 2a. Finally, circulation Type3 denotes that the stations are located south
of the anticyclonic anomaly, and the intensity and range of the anticyclonic anomaly are
significantly stronger than the other two patterns. The differences between the types imply that
severe $HD_{EC}$ may be related to different causes.

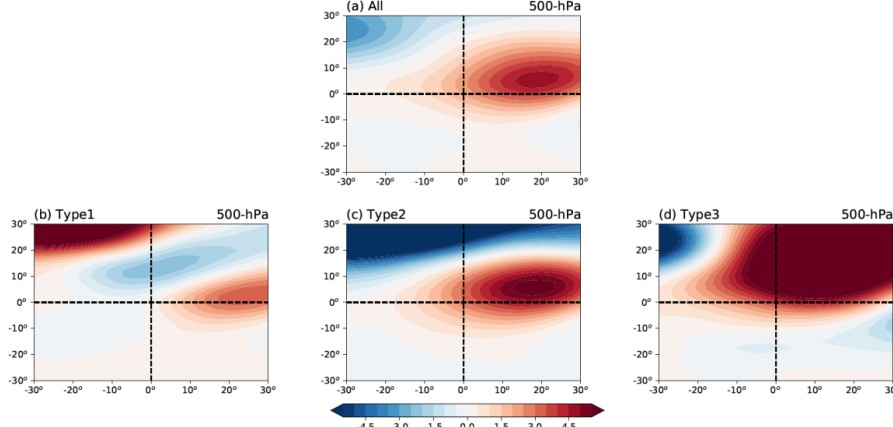


**Figure 2.** (a) Composite anomalies of GPH at 500-hPa (units: gpm) during all severe $HD_{EC}$ in 853
stations. (0°, 0°) represents the location of stations. (b), (c), and (d) are same as (a) but for three sub-
types.

For each station, when the probability of a certain circulation type is greater than the sum of

the other two types, we define this type as the dominant type of the station. Figure 3 shows the
leading circulation types of severe $HD_{EC}$ for 853 stations and the weighted probability density
distribution of three circulation types (the weight of each station is the probability of the
corresponded dominant type occurring at the station). Stations dominated by the circulation Type1
are mainly distributed in the Yangtze River valley (YRV). The stations dominated by the circulation
Type2 cover almost the whole EC, with two centers in South China (SC) and Beijing-Tianjin-Hebei
region. The stations dominated by the circulation Type3 are mainly located in Northeast China
(NEC). These results suggest significant differences in the circulation patterns of severe haze in
different regions of EC.

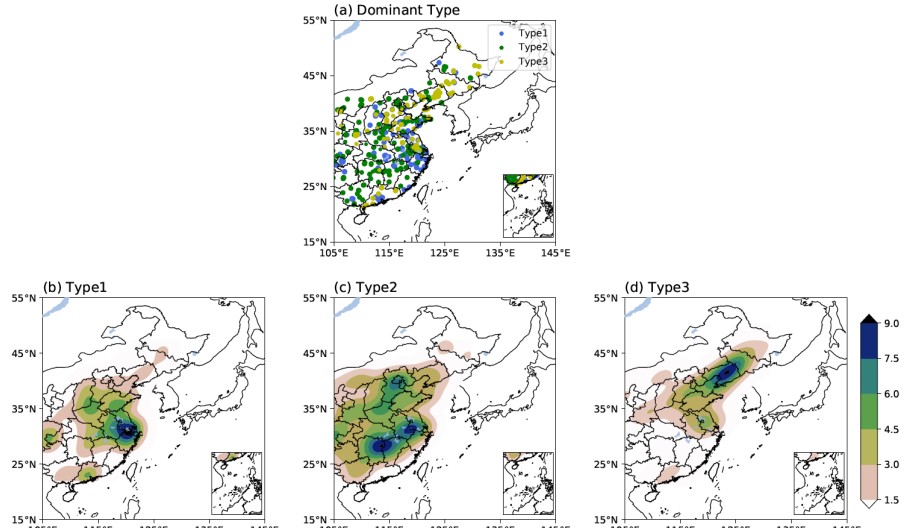

**Figure 3.** (a) The leading synoptic circulation type of severe HD$_{EC}$ for 853 stations. Weighted probability density distribution of stations dominated by (b) Type1, (c) Type2, and (d) Type3.

## 4. Comparison of different circulation types associated with severe HD$_{EC}$

Figure 4a, b, and c show the composite anomalies of circulation Type1 at 500-hPa and 850-hPa. The circulation Type1 is associated with the upper troposphere's wave-train structure of "- + -". Unlike previous studies (Zhong et al., 2019; Wang and Zhang, 2020), there are no significant anticyclonic anomalies in the mid-troposphere over YRV, but with substantial north wind component in the lower troposphere over northern China. The TIP, sinking movement, and RH anomalies over the YRV are weak (Figure 4d, e, f). Therefore, it can be inferred that it is not the local circulation anomalies that promote the formation and accumulation of haze pollution, but the regional haze transportation caused by the north wind component anomalies that leads to the severe haze in the YRV.



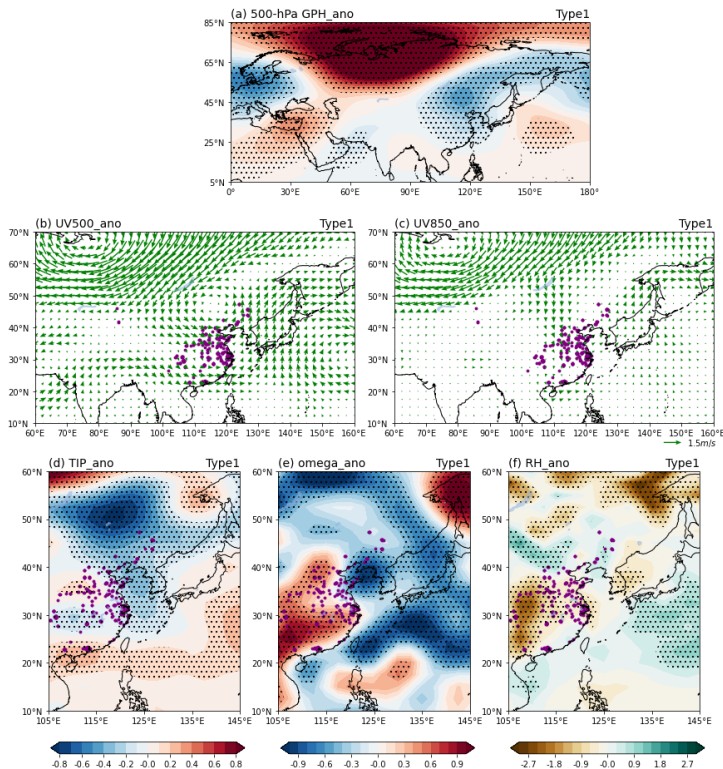


**Figure 4.** Composite anomalies of (a) GPH at 500-hPa (unit: gpm), horizontal wind (unit: m s$^{-1}$) at
(b) 500-hPa and (c) 850-hPa, (d) TIP (unit: K), (e) omega (unit: $10^{-2}$ Pa s$^{-1}$), and (f) RH (unit: %)
for circulation Type1. Dotted areas are statistically significant at the 95% confidence level. The
purple dots represent the stations dominated by circulation Type1.

To further explore the relationship between Type1 severe $HD_{EC}$ and north wind component
anomalies, we present the evolution of $PM_{2.5}$ concentration variations ($PM_{2.5}$ concentration on $Day_i$
minus that on $Day_{i-1}$) from -3 days to 2 days of Type1 severe $HD_{EC}$ occur (Figure 5a, b, c, d, e) and
the corresponding horizontal wind variations at 500-hPa (Figure 5 f, g, h, i, j). $PM_{2.5}$ concentration
tends to increase at first and then dissipate showing an obvious transportation process from north to
south. Accordingly, the horizontal wind changes from anticyclonic anomalies to cyclonic anomalies,
with the south wind turning to the north wind. Here we average the $PM_{2.5}$ concentration variations
in Figure 5a, b, c, d, e, and meridional wind variations in Figure 5f, g, h, i, j along latitudes (Figure
5k, l, m, n, o). The result shows that $PM_{2.5}$ concentration gradually increased from north EC to south



EC and began to decrease after severe HD$_{EC}$ occurred. With the variation in PM$_{2.5}$ concentration,
the south wind in the north EC gradually weakens and turns to the north wind when severe HD$_{EC}$
occurs. With the dry and cold air from the north invading southward, the haze dissipates rapidly,
and EC can maintain high air quality weather. Therefore, although circulation Type1 will lead to
severe haze in YRV, its circulation anomalies do not match the conditions to maintain haze pollution.

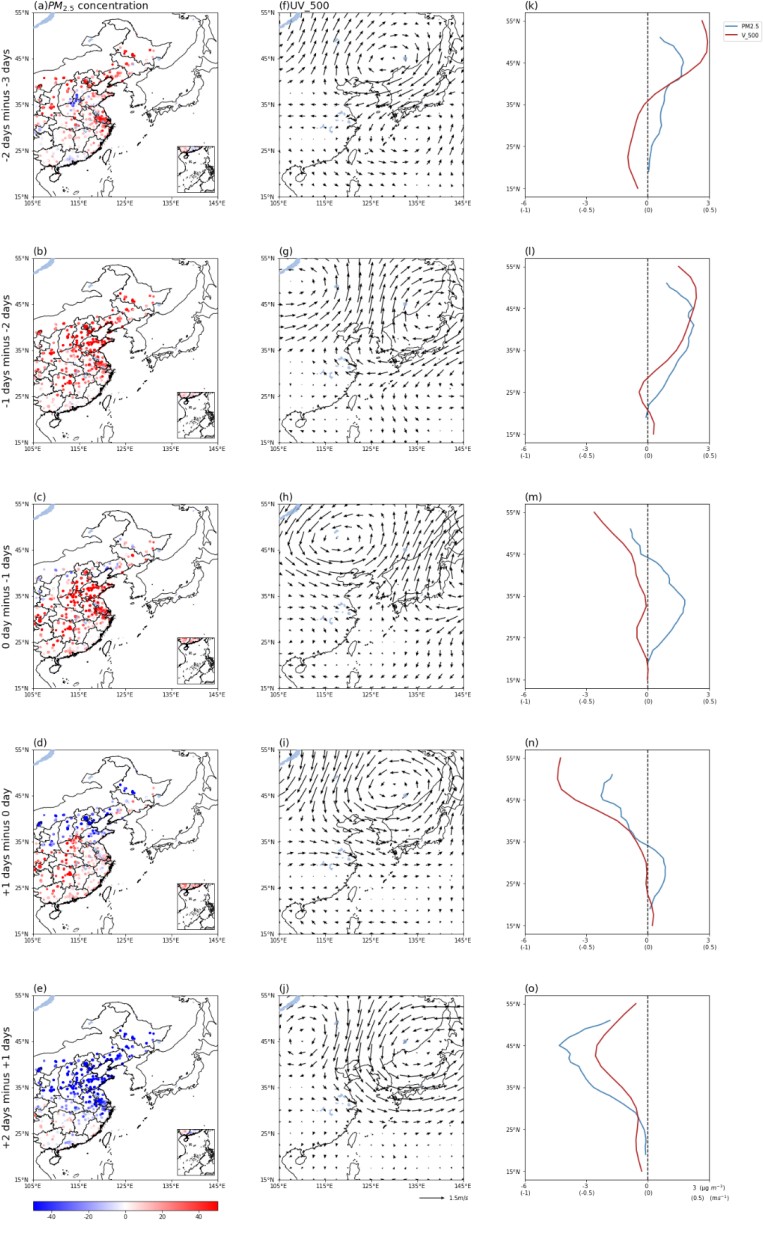




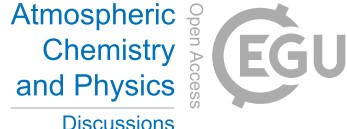
**Figure 5.** Composite anomalies of (a-e) the spatial distribution of $PM_{2.5}$ concentration (unit: µg m⁻
³) from -3 days to 2 days related to Type1 severe $HD_{EC}$ occur and (f-j) the corresponding horizontal
wind (unit: m s⁻¹) at 500-hPa. (k-o) shows the zonal averaged $PM_{2.5}$ concentration variations (unit:
µg m⁻³) and meridional wind variations (unit: m s⁻¹) in the range of 15-55°N, 105-135°E.

During the occurrence of circulation Type2, there was an anticyclonic anomaly with a quasi-
barotropic structure over Northeast Asia, and the EC was located in the southwest of the anticyclone
(Figure 6a, b, c). The significant positive TIP, sinking movement, and positive RH anomalies control
the region over EC (Figure 6d, e, f). With the increase in TIP and the warm and humid air from the
sea transports to the EC, the horizontal and vertical dispersion of pollutants was restrained, while
higher surface RH exacerbated the formation of particulates. Such circulation anomalies are
beneficial for the formation and maintenance of haze pollution.

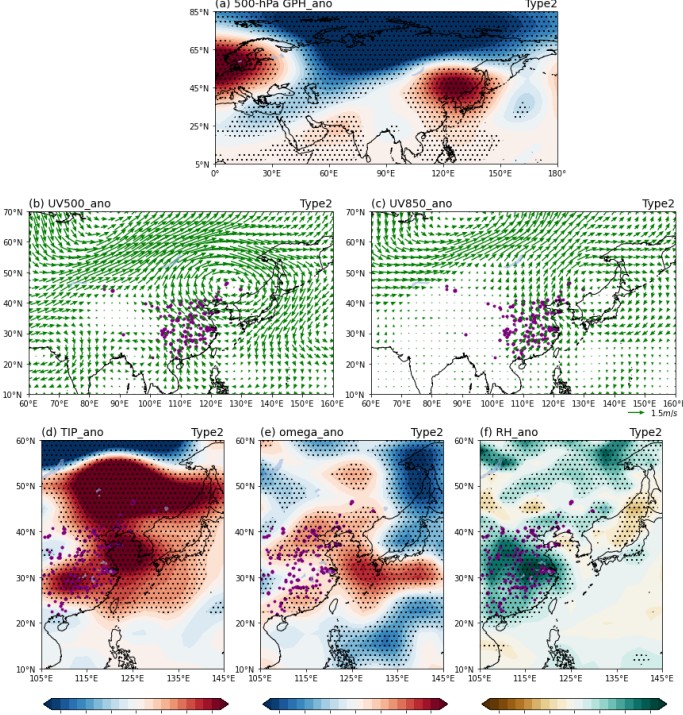


**Figure 6.** Composite anomalies of (a) GPH at 500-hPa (unit: gpm), horizontal wind (unit: m s⁻¹) at
(b) 500-hPa and (c) 850-hPa, (d) TIP (unit: K), (e) omega (unit: 10⁻² Pa s⁻¹), and (f) RH (unit: %)
for circulation Type2. Dotted areas are statistically significant at the 95% confidence level.

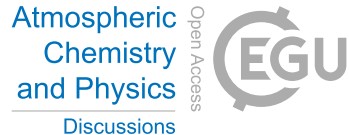

Here we investigate the dynamic mechanism of the circulation Type2 by compositing the GPH
and WAF anomalies in the upper troposphere. The circulation anomalies show two quasi-zonal wave
trains over the mid-high latitudes. The one is characterized by a '-+-+' pattern of GPH anomalies
from the south of Greenland across Siberia to Northeast China, with positive GPH anomalies in the
second and fourth centers. Such anomalies are similar to the positive phase of EA/WR
teleconnection, which can strengthen stable weather conditions over EC (Wu et al., 2016; Yin and
Wang, 2016) by causing weak wind speed, higher RH, and strong TIP (Niu et al., 2010; Ding and
Liu, 2014; Cai et al., 2017). Figure 7c shows the correlation coefficients between $PM_{2.5}$
concentration during the occurrence of circulation Type2 and the EA/WR index (The EA/WR index
was computed by the NOAA climate prediction center according to the rotated principal component
analysis used by Barnston and Livezey (1987)). The results show significant positive correlations
between the two in north EC and weak negative correlations in south EC. However, the circulation
Type2 caused the severe $HD_{EC}$ for almost the whole EC, which is not completely consistent with
the results of Figure 7c. Therefore, we speculate that the other wave-train may lead to haze pollution
in south EC.

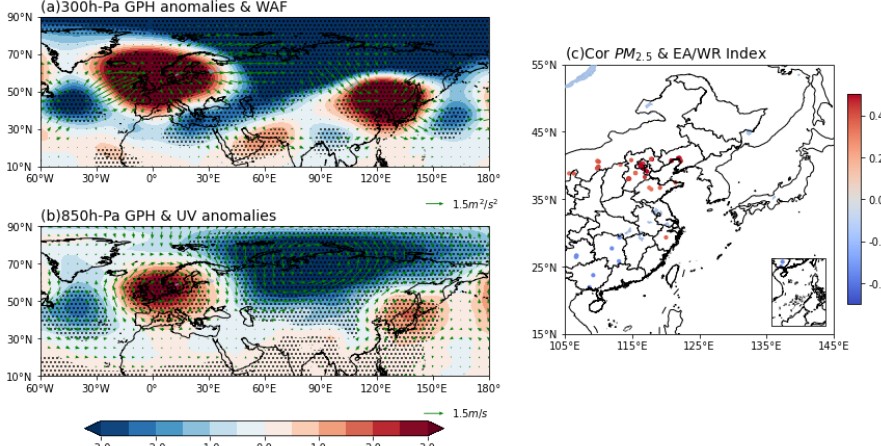


**Figure 7.** Composite anomalies of (a) GPH (shading; gpm) and WAF (vectors; $m^2 s^{-2}$) at 300-hPa,
and (b) GPH (shading; gpm) and horizontal wind (vectors; $m s^{-1}$) at 850-hPa for Type2. Dotted areas
are statistically significant at the 95% confidence level. (c) Correlation coefficients between Type2
$PM_{2.5}$ concentration and EA/WR index.



It can be found that the second wave-train reaches EC from Europe along with southern Asia,
forming a '+- + - +' pattern of GPH anomalies. The formation of such a wave-train is closely related
to the winter East Asia subtropical jet (EASJ) (Xiao et al., 2016; An et al., 2020; Zhang et al., 2022).
Here we use an Empirical orthogonal function (EOF) analysis of zonal wind from 1980 to 2021 to
determine the leading modes of winter EASJ (Xiao et al., 2016). The variance of the first mode
(EOF1) accounts for 57.4% of the total variance and indicates the intensity of EASJ (Figure 8a),
which could significantly affect the haze pollution in EC (An et al., 2020; Zhang et al., 2022).
The correlation coefficients between daily $PM_{2.5}$ concentration and the first principle component
(PC1_jet) during the occurrence of circulation Type2 is shown in Figure 8b, which has significant
positive correlations in south EC and negative correlations in north EC. It indicates that the
circulation Type2 may cause severe haze pollution in most areas of EC under the joint affection of
EA/WR teleconnection and winter EASJ. The results suggested that when discussing the impact of
an anticyclonic anomaly in Northeast Asia on haze pollution in EC, we should comprehensively
consider the joint affection of signals from high and middle latitudes.

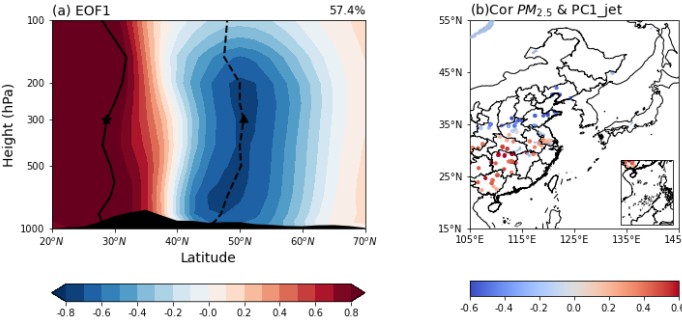


**Figure 8.** (a) The first EOF mode of zonal wind (EOF1, m s$^{-1}$) averaged from 60°E to 160°E in
NDJF. The star and circular at 300-hPa denote the subtropical jet and polar-front jet cores,
respectively. The zonal mean orography is dark-shaded. (b) Correlation coefficients between Type2
$PM_{2.5}$ concentration and PC1_jet.

Compared with circulation Type2, the range and intensity of anticyclonic anomalies in Northeast
Asia circulation Type3 are more robust, and the location is more northerly (Figure 9a). Such
circulation anomalies lead to southeasterly wind anomalies at 850-hPa, strong TIP, and abundant
moisture that induces severe haze over NEC (Figure 9d, f). In addition, the ascending motion over



the south EC and the descending motion over the Beijing–Tianjin–Hebei region and NEC formed
meridional circulation cell anomalies (Figure 9e), which are conducive to the accumulation of
severe $HD_{EC}$ over the NEC.

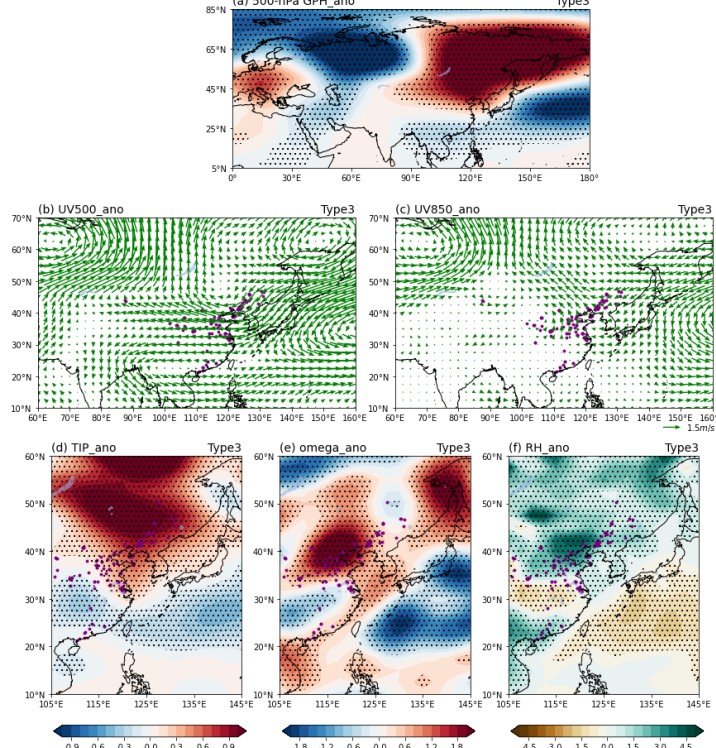


**Figure 9.** Composite anomalies of (a) GPH at 500-hPa (unit: gpm), horizontal wind (unit: m s$^{-1}$) at

(b) 500-hPa and (c) 850-hPa, (d) TIP (unit: K), (e) omega (unit: $10^{-2}$ Pa s$^{-1}$), and (f) RH (unit: %)

for circulation Type3. Dotted areas are statistically significant at the 95% confidence level.


In winter, the anticyclonic anomalies over the Okhotsk Sea are usually related to atmospheric
blocking (Yun and Yoo 2019; Fang et al., 2020; Hwang et al., 2022). Therefore, we calculated the
daily atmospheric blocking introduced in section 2.3 to investigate the relationship between Type3
severe $HD_{EC}$. Figure 10 shows that when Type3 severe $HD_{EC}$ occurs, the $PM_{2.5}$ concentration
increases with the blocking anomalies in the high-latitudes build-up, dissipating with the blocking
anomalies crash. The blocking anomalies strengthen the TIP and sufficient RH in the lower
atmosphere (Figure 11), causing severe $HD_{EC}$ in NEC.



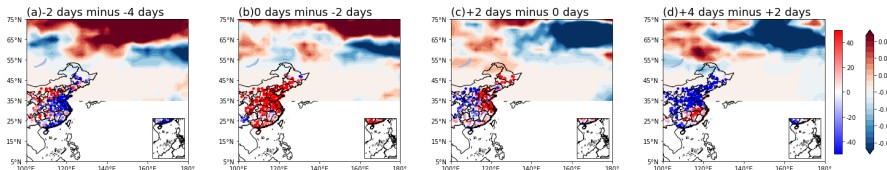

**Figure 10.** Composite anomalies of (a-d) the spatial distribution of PM$_{2.5}$ concentration variations

(unit: μg m$^{-3}$) and blockings from -4 days to 4 days related to Type3 severe HD$_{EC}$ occur.

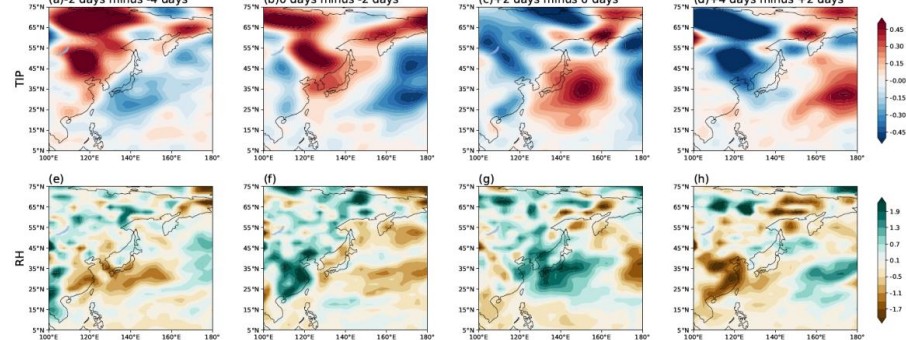

**Figure 11.** Composite anomalies of (a-d) TIP variations (unit: K) and (e-h) RH variations (unit: %)

from -4 days to 4 days related to Type3 severe HD$_{EC}$ occur.

Based on the previous studies and the differences in the influence range of the three circulations

types in this study, we divided the EC into NEC (40°N-54°N, 105°E -135°E), North China (NC;

33°N-40°N, 105°E -122°E), the YRV ( 27°N-33°N, 105°E -122°E), and SC (22°N-27°N, 105°E -

122°E) to analyze the temporal characteristics of three HD$_{EC}$ types in different subregions of EC

(Figure 12a). Figure 12b, c, d, and e display the annual regional averaged frequency of the three

HD$_{EC}$ types in the four subregions. The results show that severe haze pollution mainly occurs in NC

and less in SC. The frequency of severe haze generally shows a downward trend in the four

subregions.


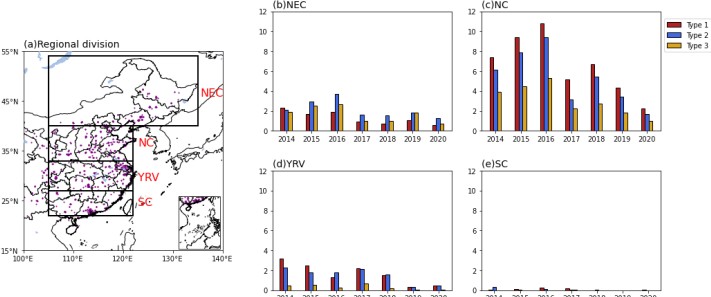

**Figure 12.** (a) The four subregions of EC. The purple dots are the stations. (b-e) Frequency of three types of cool season severe $HD_{EC}$ in NEC, NC, YRV, and SC.

We further calculated the proportion of the frequency of each circulation type in the total annual severe haze frequency in the four subregions (Figure 13). For NEC, the proportion of the three circulation types is almost equal. It should be noted that the proportion of the circulation Type3 is much larger than that in the other three subregions. In NC, the proportion of the circulation Type1 is more than 40%, while the proportion of the circulation Type3 is about 20%. For YRV, circulation Type1 and Type2 lead the severe haze pollution. There are relatively few severe haze pollution in SC. Therefore, the dominant circulation type in SC has strong interannual variation and is hardly affected by the circulation Type3. Overall, on the weather scale, the $HD_{EC}$ is affected by a variety of synoptic circulations, and the areas affected by each synoptic circulation are also different.

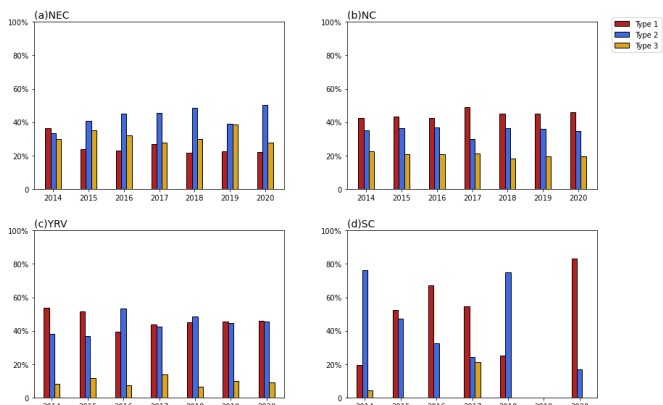

**Figure 13.** Annual percentage of the three types of severe $HD_{EC}$ in (a) NEC, (b) NC, (c) YRV, and (d) SC.



### 5. Conclusions and discussion


In this study, the Hierarchical Clustering Algorithm was used to investigate three dominant
circulation types that could lead to severe $HD_{EC}$. We cluster the circulations over the stations in EC
on the severe haze days from 2014 to 2021, which eliminates the interference of the circulations of
non-severe haze days on the cluster results. The results show that three dominant circulation types
associated with severe $HD_{EC}$ are obtained, which are mainly characterized by a local anticyclonic
anomaly but also present obvious spatial variation on large scale circulations. The circulation Type1
with wave-train structure of "-+-" in the upper troposphere mainly causes severe haze pollution in
the YRV through the low-level north wind anomalies over NC. Although the sinking movement,
TIP, and RH anomalies over the YRV are weak or not significant, the regional haze transportation
leads to the severe haze in the YRV. The circulation Type2 is characterized by two quasi-barotropic
Rossby wave trains at 300-hPa, which may be stimulated and sustained by the joint affection of
EA/WR teleconnection and the winter EASJ. One travels from the south of Greenland across Siberia
to NEC, forming a '-+-+' pattern of GPH anomalies, and the other travels from Europe along with
southern Asia, forming a '+-+-+' pattern of GPH anomalies, which led an anticyclonic over
northeastern Asia and conducive to the accumulation of haze. The circulation Type3 is characterized
by blocking anomaly over Okhotsk Sea, which influences the severe $HD_{EC}$ over NEC with
southeasterly wind at 850-hPa, strong TIP, and abundant moisture. The temporal characteristics of
three circulation types in NEC, NC, YRV, and SC were further analyzed. The result shows that on
the synoptic scale, $HD_{EC}$ is affected by various synoptic atmospheric circulations, and the regions
affected by each synoptic atmospheric circulation are also different.
The study shows that circulation patterns and key systems that contribute to severe $HD_{EC}$ are
complex and diverse revealing the dominant circulation patterns of severe haze in different regions
of EC. These three dominant atmospheric circulation patterns could be potentially used to establish
severe winter haze prediction models for different regions of EC (e.g., project the future variations
of severe haze in different regions of EC by identifying similar circulation patterns through machine
learning or regression fitting). Due to the limitation of data, it is difficult to carry out the work of
circulation classification over a longer period. Therefore, whether there is an interannual or
interdecadal connection between the dominant circulation types of severe haze and its key
circulation system needs further investigation. This study shows that different circulation types may
lead to severe haze in different regions of EC, and further studies are needed to investigate whether



there are differences in persistence or intensity among them.
**Data availability**

The Daily $PM_{2.5}$ concentrations for 935 meteorological stations in China are collected by the

China National Environmental Monitoring Centre archive at: https://quotsoft.net/air/ (last access:
16 May 2022). Daily mean meteorological data are obtained from the NCEP/NCAR reanalysis data
archive at: https://psl.noaa.gov/data/gridded/data.ncep.reanalysis.pressure.html (last access: 16
May 2022, NCEP/NCAR, 2022). The monthly EA/WR Index (CPC, 2022) can be downloaded from
NOAA's Climate Prediction Center: http://www.cpc.ncep.noaa.gov/data/teledoc/telecontents.shtml
(last access: 16 May 2022).
**Competing interests**

The authors declare that they have no conflict of interest.

**Author contributions**

SZ and GZ put forward the conception of this paper, TW improved the research and manuscript.

SZ, XY and IV performed research. SZ wrote the manuscript with contributions from all co-authors.
**Acknowledgments**

This research is supported by the National Natural Science Foundation of China (42175035 and

42077192) and the Postgraduate Research & Practice Innovation Program of Government of Jiangsu
Province (KYCX22_1162).

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
