# Peer review of "Three dominant synoptic atmospheric circulation"

_Atmospheric Chemistry and Physics, 2022_

## Referee Comment (RC2)

Three dominant synoptic atmospheric circulation patterns influencing severe winter haze in eastern China

The synoptic atmospheric circulation patterns influence haze pollution. This study employed the Hierarchical clustering classification algorithm to track haze days in the cool season over eastern China under three atmospheric circulation patterns. This manuscript also analyzed the the possible mechanisms of each circulation pattern. The paper is well written, but the methods seems to be flawed. The following concerns are addressed.

**General comments:**

1. In the introduction section, the authors only introduced weather conditions and circulation system favoring haze pollution, and further pointed out "there is still a lack of research on the dominant circulation patterns of severe $HD_{EC}$". However, there are many previous studies on circulation patterns conductive to air pollution in China. The authors did not introduce previous related studies and clear clarify the difference and the novel of the current study.

References:

*Gong, S., Liu, Y., He, J., Zhang, L., Lu, S., & Zhang, X. (2022). Multi-scale analysis of the impacts of meteorology and emissions on PM2. 5 and O3 trends at various regions in China from 2013 to 2020 1: Synoptic circulation patterns and pollution. Science of The Total Environment, 815, 152770.*

*Chang, W., & Zhan, J. (2017). The association of weather patterns with haze episodes: Recognition by PM2. 5 oriented circulation classification applied in Xiamen, Southeastern China. Atmospheric Research, 197, 425-436.*

*Liu, N., Zhou, S., Liu, C., & Guo, J. (2019). Synoptic circulation pattern and boundary layer structure associated with PM2.5 during wintertime haze pollution episodes in Shanghai. Atmospheric Research, 228, 186-195.*

*Liao, Z., Xie, J., Fang, X., Wang, Y., Zhang, Y., Xu, X., & Fan, S. (2020). Modulation of synoptic circulation to dry season PM2.5 pollution over the Pearl River Delta region: An investigation based on self-organizing maps. Atmospheric Environment, 230, 117482.*

*Sun, Y., Niu, T., He, J., Ma, Z., Liu, P., Xiao, D., ... & Yan, X. (2020). Classification of circulation patterns during the formation and dissipation of continuous pollution weather over the Sichuan Basin, China. Atmospheric Environment, 223, 117244.*

*Li, J., Liao, H., Hu, J., & Li, N. (2019). Severe particulate pollution days in China during 2013–2018 and the associated typical weather patterns in Beijing-Tianjin-Hebei and the Yangtze River Delta regions. Environmental Pollution, 248, 74-81.*

*Yang, Y., Zhou, Y., Li, K., Wang, H., Ren, L., Zeng, L., ... & Liao, H. (2021). Atmospheric circulation patterns conducive to severe haze in eastern China have shifted under climate change. Geophysical Research Letters, 48(23), e2021GL095011.*

2. The $HD_{EC}$ definition (PM$_{2.5}$ ≥150 μg m$^{-3}$) would lead to underestimation of the role

of circulation pattern in recent years. Considering the implementation of the Air Pollution Prevention and Control Action Plan since 2013, there is a decreasing rate of 7% in PM$_{2.5}$ concentrations in eastern China. The impact of emission change on PM$_{2.5}$ should be removed before the HD$_{EC}$ definition.

3. The process about classification of circulation does not seem convincing. The circulation patterns are identified not in fixed region, but a rectangular region of 30 degree with each station as the center. However, China is a vast country with a diverse climate, and uneven spatial patterns of meteorological conditions and air pollutants. The classification may be not reliable. For example, Figure 2(c) shows that the stations with severe haze are located in the southwestern parts of the anticyclonic anomaly center. But the locations of anticyclonic anomaly are different for stations in north and south China. It is recommended to first divide eastern China into four subregions like Figure 12 (a), and then begin to classify key circulation patterns in each region.

4. The authors only cluster the circulation anomalies of days with severe HD$_{EC}$. Would such data processing be robust? It is recommended to add box plots of the daily PM$_{2.5}$ concentrations under each circulation type. Please add the frequency of occurrences for each type in Figure 2 also.

---

## Author Comment (AC1)

**Response to the Comments**

Dear reviewer,

We thank you so much for taking time to enhance the quality of our paper. We have revised the manuscript, and changes are shown with red color in the revised manuscript. Below are our responses to the reviewers' comments. All reviewers' comments are in black, while the authors' responses are in blue. And all revisions in the revised manuscript are highlighted in red color.

The physical mechanism of severe winter haze in eastern China has been revealed in this work. Three dominant atmospheric circulation patterns effecting the haze occurrence have been clustered. The paper is generally well written and recommended for publication after addressing the following specific comments.

1. Air pollution mainly occurred in the atmospheric boundary layer, which is usually under 2km above the surface. While, the 500hPa geopotential height anomalies are used for circulation clustering. Could you please give more details about the reason for selecting the 500hPa data?

**Response:** Previous studies have shown that the upper-level circulation represented by 500-hPa geopotential height anomalies play an important role in the generation and accumulation of haze (Wang et al., 2015; Yin and Wang, 2017; Zhong et al., 2019). On the one hand, the upper-level circulation can affect the haze through meteorological factors such as thermal inversion potential and vertical movement; on the other hand, the upper-level circulation can also affect the haze by regulating the near surface circulation. Secondly, we focus on a large spatial scale circulation anomaly. The near surface circulation is difficult to display obvious characteristics due to the complex terrain, so we choose to cluster the 500-hPa geopotential height anomalies.

The revised sentence is as follows (see lines 135-141): Secondly, the circulation samples selected are not in a fixed region, but the rectangular regions of the same size centered on each station with severe haze. Since the upper-level circulation represented by 500-hPa geopotential height anomalies play an important role in the generation and accumulation of haze (Wang et al., 2015; Yin and Wang, 2017; Zhong et al., 2019), the GPH anomalies at 500-hPa in a rectangular region of 30 degrees from east, west, north, and south with each station as the center on the day of severe $HD_{EC}$ were taken as the samples to perform HCA.

**References:**

Wang, H. J., Chen, H. P., Liu, J. P.: Arctic Sea Ice Decline Intensified Haze Pollution in

Eastern China. Atmos. Oceanic Sci., 8:1, 1-9, https://doi.org/10.3878/AOSL20140081, 2015.

Yin, Z. C., Wang, H. J.: Role of atmospheric circulations in haze pollution in December 2016, Atmospheric Chemistry and Physics, 17(18): 11673-11681, https://doi.org/10.5194/acp-17-11673-2017, 2017.

Zhong, W. G., Yin, Z. C., Wang, H. J.: The relationship between anticyclonic anomalies in northeastern Asia and severe haze in the Beijing–Tianjin–Hebei region, Atmos. Chem. Phys., 19, 5941-5957, https://doi.org/10.5194/acp-19-5941-2019, 2019.

2. The samples used in the clustering are not in a fixed region, which is a rectangular box moving with the specific observation station. This clustering method is different from the usual treatment. What is the advantage of this method?

**Response:** Due to the geographical nature of east china which is large, there may be differences in the circulation anomalies during haze pollution between the southern stations and northern stations. The circulation patterns obtained by clustering the circulation over the local area of each station are relative to the local climate statement, which is conducive to understanding the mechanism of haze formation in different regions. When we cluster the circulation anomalies in a fixed region, influence of the same circulation pattern on other stations is different due to variations in stations' location.

The sentence was revised as follows (see lines 141-143): It means that our classification results focus on the local circulation anomalies accompanied by haze, which can help us more accurately understand the impact of different local circulation patterns on different stations.

3. I would like to suggest to change a colormap for Fig. 3(a), in which the Type1 and Type 2 is hard to distinguish based on the current color

**Response:** Thanks for your suggestion. The revised figure3 is as follows:

[Figure]

**Figure 3**. Distribution of stations dominated by (a) Type1, (b) Type2, and (c) Type3 synoptic circulation pattern. Weighted probability density distribution of stations dominated by (d) Type1, (e) Type2, and (f) Type3 synoptic circulation pattern.

---

## Author Comment (AC2)

**Response to the Comments**

Dear reviewer,

We thank you so much for taking time to enhance the quality of our paper. We have revised the manuscript, and changes are shown with red color in the revised manuscript. Below are our responses to the reviewers' comments. All reviewers' comments are in black, while the authors' responses are in blue. And all revisions in the revised manuscript are highlighted in red color.

Three dominant synoptic atmospheric circulation patterns influencing severe winter haze in eastern China

The synoptic atmospheric circulation patterns influence haze pollution. This study employed the Hierarchical clustering classification algorithm to track haze days in the cool season over eastern China under three atmospheric circulation patterns. This manuscript also analyzed the the possible mechanisms of each circulation pattern. The paper is well written, but the methods seems to be flawed. The following concerns are addressed.

General comments:

1.  In the introduction section, the authors only introduced weather conditions and circulation system favoring haze pollution, and further pointed out "there is still a lack of research on the dominant circulation patterns of severe $HD_{EC}$ ". However, there are many previous studies on circulation patterns conductive to air pollution in China. The authors did not introduce previous related studies and clear clarify the difference and the novel of the current study.

    References:

    *Gong, S., Liu, Y., He, J., Zhang, L., Lu, S., & Zhang, X. (2022). Multiscale analysis of the impacts of meteorology and emissions on PM2. 5 and O3 trends at various regions in China from 2013 to 2020 1: Synoptic circulation patterns and pollution. Science of The Total Environment, 815, 152770.*

    *Chang, W., & Zhan, J. (2017). The association of weather patterns with haze episodes: Recognition by PM2. 5 oriented circulation classification applied in Xiamen, Southeastern China. Atmospheric Research , 197, 425 436.*

    *Liu, N., Zhou, S., Liu, C., & Guo, J. (2019). Synoptic circulation pattern and boundary layer structure associated with PM2.5 during wintertime haze pollution episodes in Shanghai. Atmospheric Research, 228, 186 195.*

    *Liao, Z., Xie, J., Fang, X., Wang, Y., Zhang, Y., Xu, X., & Fan, S. (2020). Modulation of synoptic circulation to dry season PM2.5 pollution over the Pearl River Delta*

*region: An investigation based on self organizing maps. Atmospheric Environment, 230, 117482.*

*Sun, Y., Niu, T., He, J., Ma, Z., Liu, P., Xiao, D., ... & Yan, X. (2020). Classification of circulation patterns during the formation and dissipation of continuous pollution weather over the Sichuan Basin, China. Atmospheric Environment, 223, 117244.*

*Li, J., Liao, H., Hu, J., & Li, N. (2019). Severe particulate pollution days in China during 2013 2018 and the associated typical weather patterns in Beijing Tianjin Hebei and the Yangtze River Delta regions. Environmental Pollution, 248, 74 81.*

*Yang, Y., Zhou, Y., Li, K., Wang, H., Ren, L., Zeng, L., ... & Liao, H. (2021). Atmospheric circulation patterns conducive to severe haze in eastern China have shifted under climate change. Geophysical Research Letters, 48(23), e2021GL095011.*

**Response:** Thank you for your comments. We reviewed and cited more references and compared the differences with this study. We have revised and supplemented the introduction (see lines 69-83 in revised manuscript): Existing studies have also investigated the synoptic circulation patterns conducive to haze pollution in different regions of China (Chang and Zhang, 2017; Li et al., 2019; Liu et al., 2019; Liao et al., 2020; Sun et al., 2020; Yang et al., 2021; Gong et al., 2022). Most of these studies produced the classification based on low-level circulation anomalies, while the upper-level circulation also play an important role in the generation and accumulation of haze (Wang et al., 2015; Yin and Wang, 2017; Zhong et al., 2019). In addition, due to the large spatial span in EC, if we assess the classification of synoptic circulation patterns in a fixed region, it may lead to different effects of the same classification pattern on different regions. Therefore, we classify the circulation anomalies with severe haze days of each station in EC respectively, and finally obtain the dominant synoptic atmospheric circulation pattern of each station. In general, the present study addresses the following scientific questions: (1) what are the synoptic atmospheric circulation patterns that dominate severe haze pollution in EC, (2) what are the differences in the action range of each circulation pattern, and what are their possible mechanisms. These

issues are addressed using a modified classification algorithm (Hierarchical Clustering Algorithm) that is more suitable for studying the classification of synoptic patterns in a large spatial range.

2. The $HD_{EC}$ definition (PM $_{2.5}$ ≥150 μg m$^{-3}$ ) would lead to underestimation of the role of circulation pattern in recent years. Considering the implementation of the Air Pollution Prevention and Control Action Plan since 2013, there is a decreasing rate of 7% in $PM_{2.5}$ concentrations in eastern China. The impact of emission change on $PM_{2.5}$ should be removed before the $HD_{EC}$ definition.

**Response:** This research focused on the relatively severe haze pollution events. Referring to some previous studies (Cai et al., 2017; Li et al., 2019; Zhong et al., 2019), we set the threshold value as 150 μg m$^{-3}$. Due to the climatological concentration of $PM_{25}$ concentration in north of EC is relative high, if we use a lower threshold to identify severe haze event, we may not be able to effectively indentify haze pollution events in the north of EC, which will mix with a large number of low concentration haze events. In addition, emissions are usually stable on a synoptic scale, and variation of meteorological factors are considered to be the main driving factors (Wang et al., 2015; Yin and Wang, 2017; Li et al., 2022). Here we use the method of relative threshold (1.5 times of standard deviation) to identify severe haze days and cluster them, and we can get similar results (Figure 1 in response letter), which shows that the research results are not sensitive to the method of defining severe haze. Subsequent work will further carefully compare the impact of emissions and meteorological factors on haze. And we have added the corresponding description in the discussion section (see lines 363-367 in revised manuscript): In addition, considering the latitude difference of $PM_{2.5}$ concentrations in EC and the decreasing of $PM_{2.5}$ concentrations due to implementation of the Air Pollution Prevention and Control Action Plan since 2013, the flexible threshold to identify haze day is suggested to use in the further studies. And we will further carefully compare the impact of emissions and meteorological factors on haze in subsequent work.

[Figure]

**Figure 1 in response letter**: (a) Composite anomalies of GPH at 500-hPa (units: gpm) during all severe HD$_{EC}$ in 853 stations. (0°, 0°) represents the location of stations. (b), (c), and (d) are same as (a) but for three sub-types.

**References:**

Cai, W. J., Li, K., Liao, H., Wang, H. J., and Wu, L. X.: Weather Conditions Conducive to Beijing Severe Haze More Frequent under Climate Change, Nat. Clim. Change, 7, 257–262, https://doi.org/10.1038/nclimate3249, 2017.

Li J., X. Hao, H. Liao, Y. Wang, W. Cai, K. Li, X. Yue, Y. Yang, H. Chen, Y. Mao, Y. Fu, L. Chen, and J. Zhu, Winter particulate pollution severity in North China driven by atmospheric teleconnections, Nature Geoscience, 15, 349-355, doi:10.1038/s41561-022-00933-2, 2022.

Li, J., Liao, H., Hu, J., & Li, N.: Severe particulate pollution days in China during 2013 2018 and the associated typical weather patterns in Beijing Tianjin Hebei and the Yangtze River Delta regions, Environmental Pollution, 248, 74 81, https://doi.org/10.1016/j.envpol.2019.01.124, 2019.

Wang, H. J., Chen, H. P., Liu, J. P.: Arctic Sea Ice Decline Intensified Haze Pollution in Eastern China. Atmos. Oceanic Sci., 8:1, 1-9, https://doi.org/10.3878/AOSL20140081, 2015.

Yin, Z. C., Wang, H. J.: Role of atmospheric circulations in haze pollution in December

2016, Atmospheric Chemistry and Physics, 17(18): 11673-11681, https://doi.org/10.5194/acp-17-11673-2017, 2017.

Zhong, W. G., Yin, Z. C., Wang, H. J.: The relationship between anticyclonic anomalies in northeastern Asia and severe haze in the Beijing–Tianjin–Hebei region, Atmos. Chem. Phys., 19, 5941-5957, https://doi.org/10.5194/acp-19-5941-2019, 2019.

3. The process about classification of circulation does not seem convincing. The circulation patterns are identified not in fixed region, but a rectangular region of 30 degree with each station as the center. However, China is a vast country with a diverse climate, and uneven spatial patterns of meteorological conditions and air pollutants. The classification may be not reliable. For example, Figure 2(c) shows that the stations with severe haze are located in the southwestern parts of the anticyclonic anomaly center. But the locations of anticyclonic anomaly are different for stations in north and south China. It is recommended to first divide eastern China into four subregions like Figure 12 (a), and then begin to classify key circulation patterns in each region.

**Response:** We presented the final composited result in the paper (Figures 4, 6, 9). According to our definition, each cluster sample makes the station with severe haze located in the center of the circulation anomalies, which ensures that each type of event has the same impact on the stations it controls. Therefore, the clustering scheme used in this paper actually takes into account the variations and motion of circulation patterns on different regions. In addition, since we cluster the 500-hPa circulation anomaly rather than the meteorological elements near the surface, the impact of terrain and other underlying surfsce factors is avoided to some extent. We can also exclude different impacts in the region by adopting the method of relative threshold, see the previous comment (2) for details. We added more descriptions of methods (see lines 152-155 in revised manuscript): Specifically, this clustering scheme can ensure that each station is located in the center of the circulation pattern when severe haze occurs, and avoid the impact of circulation pattern movement. The final composite results of the same pattern

can reflect the average statement of the current type of circulation anomaly, which is helpful to investigate its possible physical mechanism.

4. The authors only cluster the circulation anomalies of days with severe $HD_{EC}$. Would such data processing be robust? It is recommended to add box plots of the daily $PM_{2.5}$ concentrations under each circulation type. Please add the frequency of occurrences for each type in Figure 2 also.

**Response:** According to your comment, we have supplemented the corresponding figure as Figure S2. Here we present the mean value, the standard deviation of the $PM_{2.5}$ concentration and frequency of severe haze days for each types. In addition, we have added the following description in the corresponding location (see lines 189-190 in revised manuscript): In general, the stations in the north of EC are accompanied by higher $PM_{2.5}$ concentration and more haze days (Figure S2).

[Figure]

**Figure S2.** (a) Mean, (d) standard deviation of $PM_{2.5}$ concentration distribution and (g) frequency of severe haze days in cluster samples of Type1. (b), (e), (h) and (c), (f), (j) are the same as (a), (d), (g), but for Type2 and Type3, respectively.